# MetaboLabPy—An Open-Source Software Package for Metabolomics NMR Data Processing and Metabolic Tracer Data Analysis

**DOI:** 10.3390/metabo15010048

**Published:** 2025-01-14

**Authors:** Christian Ludwig

**Affiliations:** Department of Metabolism and Systems Sciences, School of Medical Sciences, College of Medicine and Health, University of Birmingham, Birmingham B15 2TT, UK; c.ludwig@bham.ac.uk

**Keywords:** NMR, metabolomics, tracing, stable isotope, deep learning, Python

## Abstract

**Introduction:** NMR spectroscopy is a powerful technique for studying metabolism, either in metabolomics settings or through tracing with stable isotope-enriched metabolic precursors. MetaboLabPy (version 0.9.66) is a free and open-source software package used to process 1D- and 2D-NMR spectra. The software implements a complete workflow for NMR data pre-processing to prepare a series of 1D-NMR spectra for multi-variate statistical data analysis. This includes a choice of algorithms for automated phase correction, segmental alignment, spectral scaling, variance stabilisation, export to various software platforms, and analysis of metabolic tracing data. The software has an integrated help system with tutorials that demonstrate standard workflows and explain the capabilities of MetaboLabPy. **Materials and Methods:** The software is implemented in Python and uses numerous Python toolboxes, such as numpy, scipy, pandas, etc. The software is implemented in three different packages: metabolabpy, qtmetabolabpy, and metabolabpytools. The metabolabpy package contains classes to handle NMR data and all the numerical routines necessary to process and pre-process 1D NMR data and perform multiplet analysis on 2D-^1^H, ^13^C HSQC NMR data. The qtmetabolabpy package contains routines related to the graphical user interface. **Results:** PySide6 is used to produce a modern and user-friendly graphical user interface. The metabolabpytools package contains routines which are not specific to just handling NMR data, for example, routines to derive isotopomer distributions from the combination of NMR multiplet and GC-MS data. A deep-learning approach for the latter is currently under development. MetaboLabPy is available via the Python Package Index or via GitHub.

## 1. Introduction

NMR-based metabolomics is a cornerstone of the toolbox available to analytical chemists to investigate distortions in metabolic networks in health and disease. For this purpose, 1D-^1^H NMR spectroscopy is still one of the most frequently used quantitative techniques available. However, to extract reliable and significant information from such NMR spectra, a robust and reliable data processing and pre-processing workflow is essential. Such a workflow must include consistent phase correction for a series of densely populated 1D-^1^H NMR spectra. This is often difficult to achieve because of baseline distortions, especially when the NMR instrumentation includes a CryoProbe. While the baseline distortion exacerbates the task of consistently and precisely phase correcting the NMR spectra, a baseline correction requires properly phased NMR spectra first. The workflow must then include a baseline correction algorithm without distorting NMR signal intensities and shapes. In particular, urine samples suffer from pH variations between different samples, which leads to local chemical shift variations of a subset of NMR signals. Therefore, the data processing workflow needs to include the possibility of segmental alignment to minimise artefacts caused by such pH differences.

To optimally prepare the NMR spectra for univariate and multivariate data analysis, the workflow needs to include the possibility of excluding baseline regions, linear [1] and non-linear [2] scaling algorithms, bucketing of data, noise filtering, and the export of data such that the pre-processed NMR spectra can be imported into publicly available statistical data analysis tools such as MetaboAnalyst [3] https://www.metaboanalyst.ca (accessed on 6 January 2025). While an intuitive graphical user interface (GUI) facilitates such NMR data processing and makes it possible for non-experts to perform these tasks, it must also be possible to integrate the data processing workflow into non-graphical workflows.

In addition to targeted or untargeted metabolomics studies, metabolic tracing is now a major tool for investigating metabolic changes in great detail. Let us assume that the metabolic pathways affected by a disease are known. In this case, it is possible to feed cells, organs, or even whole organisms with metabolic precursors of that pathway enriched with rare stable isotopes such as ^13^C and to follow how these precursors are metabolised. While gas chromatography-coupled mass spectrometry (GC-MS) is an important analytical tool with which to investigate samples obtained after exposing cells, etc., for some time (typically between 24 and 72 h), GC-MS data lacks atomic resolution. With NMR spectroscopy data, on the other hand, it is possible to obtain tracing information in downstream metabolites at atomic resolution. While GC-MS data contain information about ^12^C and ^13^C incorporation into metabolites, NMR spectroscopy is blind to ^12^C and requires overall scaling of ^13^C incorporation. This can be achieved via the preparation of two separate samples, one using unlabelled metabolic precursors (e.g., glucose) and another sample using ^13^C enriched metabolic precursors (e.g., [1,2-^13^C] glucose). While this method solves the NMR scaling problem, it will limit the application of metabolic tracing to samples derived from cell culture. A combination of GC-MS data with NMR spectroscopy data solves this problem [4,5]. Therefore, software designed to analyse metabolic tracing data should ideally include algorithms to analyse NMR spectroscopic tracing data and integrate these with GC-MS-derived data.

## 2. Materials and Methods

MetaboLabPy is an entirely new software package that is not based on any previous software. It is implemented in Python and available as free and open-source software (FOSS) from GitHub (https://github.com/ludwigc/metabolabpy accessed on 5 January 2025, https://github.com/ludwigc/qtmetabolabpy accessed on 5 January 2025, https://github.com/ludwigc/metabolabpytools accessed on 5 January 2025) and the Python Package Index (PyPI; https://pypi.org/project/metabolabpy accessed on 5 January 2025, https://pypi.org/project/qtmetabolabpy accessed on 5 January 2025, https://pypi.org/project/metabolabpytools accessed on 5 January 2025). The objectives for developing MetaboLabPy were as follows:To create software with a user-friendly GUI to allow non-expert users to process and analyse NMR-based metabolomics and metabolic tracing data.To provide software enabling spectral data processing from scratch, i.e., not relying on other—potentially commercial—software packages for NMR data processing.To enable scripted batch-processing of large series of NMR spectra.To provide a robust and easy-to-use workflow for data pre-processing of NMR-based metabolomics data.To enable exporting pre-processed data to various statistical analysis software packages such as MetaboAnalyst [3], Batman [6], rDolphin [7], Chenomx (https://www.chenomx.com/ accessed on 5 January 2025), or other software packages that may use Excel files as data input.To enable non-expert users to process, assign, and analyse ultra-high resolution 2D HSQC NMR spectra.To assist HSQC assignment with a GUI and a library of over 100 metabolites with chemical shift information derived from online databases such as HMDB [8] and BMRB [9].To analyse tracing data by providing a graphical interface for semi-automated line shape analysis.To derive isotopomer distributions from a combination of NMR HSQC and GC-MS data [4,10].To enable users to use the software either via the GUI or without a GUI, for example, in a Jupyter Notebook environment.To include a plot editor enabling the creation of publication-quality plots of processed NMR spectra.

The software implementation has been split into three different software packages. The metabolabpy package contains classes to handle NMR data and all numerical routines to process and pre-process 1D NMR data and perform multiplet analysis on 2D-^1^H, ^13^C HSQC NMR data. The qtmetabolabpy package contains routines related to the graphical user interface. PySide6 is used to produce a modern and user-friendly graphical user interface. The metabolabpytools package contains valuable routines not specific to handling NMR data and deriving isotopomer distributions from the combination of NMR multiplet and GC-MS data.

The package metabolabpy contains the core of all available routines for data input and output as well as computational procedures. Several classes are defined in this package: The main class for the software is nmrData. This class contains the data structures to hold all information necessary to process, reference, and analyse NMR spectra. This includes classes representing acquisition data (class: acqPars), processing options (class: procPars), display options, such as colour, number of contour lines, etc. (dispPars), as well as classes to parse acquisition and processing parameter files for Bruker and Varian file formats (classes: acqRegEx, acqProcparRegEx, procRegEx, procProcparRegEx). Although, the user usually does not interact with these classes directly. MetaboLabPy implements the nmrDataSet class to enable handling of series of NMR spectra. This class also contains routines that are applied to series of NMR spectra, such as metabolomics data pre-processing options.

The qtmetabolabpy package contains the qtmetabolabpy class, which defines the graphical user interface and includes routines that can be entered at the MetaboLabPy command line. This class also implements interactive plotting and phase correction routines. All functions that are available via the menus have associated keyboard shortcuts.

MetaboLabPy takes advantage of numerous other open-source software packages available in Python. The required packages for MetaboLabPy to work are as follows: NumPy, SciPy, MatplotLib, Pandas, Numba, scikit-learn, openpyxl, xlsxwriter, multiprocess, PyBaselines, darkdetect, pywavelets, pygamma, mat73, PySide6, pyautogui, qtmodern, xlsxwriter, pypdf2, and if the software is to be used from within a Jupyter-notebook, Jupyter. All these packages are available on PyPI and will be installed automatically when installing the PyPI packages of MetaboLabPy. As the pygamma package is only easily available for Python 3.9, we recommend installing Python version 3.9.6.

Two sets of samples were produced to test the performance of automatic phase correction and segmental alignment procedures. The first sample set consisted of five technical replicates of a metabolite mixture (50 mM urea, 0.78 mM creatinine, 0.68 mM alanine, 0.23 mM isoleucine, and 0.5 mM TMSP in 100 mM phosphate buffer at pH 7.0). The second set of samples contained four *Daphnia magna* extracts. The organisms were extracted using a CHCl_3_/methanol/water mixture in the ratio 1:1:1. The polar fraction was dried overnight in a SpeedVac and dissolved in 100 mM phosphate buffer at pH 7.0, containing 0.5 mM TMSP.

The 1D-^1^H NMR spectra were acquired using the Bruker standard sequence noesygppr1d. For the metabolite mix, 32 transients were recorded after four steady-state scans. The relaxation delay was set to 4 s. Spectra were acquired using a 600 MHz Avance III IVDr system (Bruker, Coventry, UK) equipped with a room-temperature inverse broadband probe. For the *Daphnia* extract samples, 128 transients were recorded after eight steady-state scans. The relaxation delay was set to 3 s. The instrument used to acquire the NMR spectra was a 600 MHz Avance III Bruker NMR system (Bruker, Coventry, UK) equipped with an inverse, 1.7 mm CryoProbe.

## 3. Results

### 3.1. Implementation and Class Description

The main package for the software is the metabolabpy package. This package contains several classes to implement the input and output of NMR data and all computational routines. Appendix A shows a schematic plot of the class structure of MetaboLabPy. The main class for a single NMR spectrum is NmrData. This class contains numpy array’s to store the free induction decay, as well as the NMR spectrum. All computational routines operating on a single spectrum are implemented here, including Bruker or Varian data. This class imports a variety of other classes implemented as part of the software. These are as follows:AcqPars: This class contains all acquisition parameters. It imports two more classes, AcqRegEx and AcqProcparRegEx, which contain regular expressions used to extract acquisition data information for Bruker and Varian file formats, respectively.ProcPars: Similar to AcqPars, this class imports ProcRegEx and ProcProcparRegEx, which contain regular expressions to extract processing information for Bruker and Varian file formats, respectively.NmrpipeData: This class implements procedures to read in NmrPipe processed data.DispPars: In this class, information regarding the visual appearance of the NRM spectrum are implemented.SplineBaseline: This class implements a cubic spline baseline correction. To avoid over-correction of dense areas of the NMR spectrum with no useable baseline points, a “linear-spline” is implemented. When two adjacent baseline points are more than a set maximum number of data points apart, intermediate baseline points are simulated using a line between the two data points.NmrHsqc: This class implements multiplet analysis of 2D-^1^H, ^13^C HSQC NMR spectra. This class imports HsqcData, which provide the necessary data structures and a function to initialise the hsqcData object.Phase3: This is not a class. The file implements functions for automatic phase correction. To be able to use the numba just-in-time compiler, this needed to be implemented in a separate file.NmrConfig: This class reads and stores the MetaboLabPy configuration file and is imported into several classes.

In order to be able to read in series of NMR spectra, the NmrDataSet was implemented. This class contains everything related to procedures applied in batch processing and data pre-processing. NmrDataSet also imports NmrConfig and NmrPreProc. The latter class implements all data structures and computational routines related to data pre-processing.

The QtMetaboLabPy class finally implements the graphical user interface (GUI) and all routines that are GUI-related, e.g., interactive phase correction. The top object here is nd, which is an instance of the NmrDataSet class.

### 3.2. Data Input and Output

MetaboLabPy has its own file format, where data can be saved and later loaded into the software. This was necessary in order to include additional information, e.g., data pre-processing settings or multiplet-analysis in 2D HSQC NMR spectra. There are a variety of ways to import data. The software can read in Bruker and Varian format in 1D and 2D NMR datasets. However, owing to the departure of Varian/Agilent from the NMR spectrometer production field, the software is quite Bruker-centric, taking advantage of Bruker’s title, acquisition, and pulse program files. The 2D datasets that have been processed in NMRPipe format can be read as well.

Additionally, 1D and 2D datasets saved as a mat file in the MATLAB-based MetaboLab software (MATLAB version 2024b, the Mathworks, Natick, MA, USA; MetaboLab version 2023.0529) [11] can be imported into MetaboLabPy. All these options are available from the File menu. NMR spectra can also be exported into Bruker format for further analysis in TopSpin or Chenomx.

### 3.3. Graphical Software User Interface (GUI) and Help System

The GUI is organised as a series of tabs. The main tab categories are NMR Spectrum, HSQC Analysis, Processing Parameters, Display Parameters, Acquisition Parameters, Title File, pUlseProgram, Script, cOnsole, pLot Editor, and Help. Some tabs or other GUI elements are hidden and will be visible only under certain circumstances. For example, the checkbox allowing the appearance of the HSQC Analysis tab will only be visible if an NMR spectrum is loaded where the pulse sequence name includes the string hsqc. Likewise, the checkboxes for displaying data pre-processing options, a GUI element for peak-picking and another for spline baseline correction will only be visible for 1D NMR datasets. The design principle behind this is to offer only helpful options for the currently loaded NMR dataset. A detailed description of the GUI is available in the MetaboLabPy help, either in the Help tab of the software or online (https://ludwigc.github.io/metabolabpy, accessed on 6 January 2025). The MetaboLabPy help system contains tutorials that introduce all major aspects of how the software can be used.

### 3.4. Scripted Batch Processing

Because MetaboLabPy is implemented in Python, the software can be easily scripted using Python scripts. This can be accomplished from within the graphical user interface (GUI) or without, using the GUI, e.g., from within a Jupyter notebook. Sample scripts are available for batch processing from within the GUI, covering all basic scenarios, including reading in 1D and 2D datasets, 2D Jres-processing, and reading in a series of NMRPipe-processed NMR spectra. The MetaboLabPy software distribution also contains examples of Jupyter notebooks. This topic is also covered in the help system.

### 3.5. 1D NMR Data Processing

#### 3.5.1. Manual and Automatic Phase Correction

One of the most critical steps of NMR data processing for metabolomics is a precise and consistent phase correction as this will influence the quality of univariate and multivariate statistical data analysis. Phase correction can be performed manually from within the GUI or automated with or without the GUI. MetaboLabPy implements the algorithm described in [12]. While this algorithm usually performs well, the algorithm performs differently for different NMR spectra, even with very similar NMR signal intensities in the NMR spectra (Figure 1). Appendix A shows the automatic phase correction of small-series CryoProbe spectra of *Daphnia* extracts.

To improve the consistency of phase correction in a series of NMR spectra, MetaboLabPy defines a phase reference experiment, which can be set in the Display Parameters tab. When the software is in phase correction mode, this reference spectrum will be shown in addition to the currently selected NMR spectrum, making consistent phase correction much more straightforward.

In addition, MetaboLabPy implements a new automatic phase correction algorithm. This algorithm relies on a phase-corrected reference spectrum, which was manually or automatically phase-corrected. The algorithm starts by using the reference spectrum’s zero- and first-order phase values to phase-correct the current spectrum. The objective function to be minimised is the absolute difference between the baseline regions on the left (>10.0 ppm) and right (<−0.5 ppm) edge of the spectra. These differences are minimised using Powell’s conjugate direction method. The results are optimised zero- and first-order phase correction values for each spectrum. As shown in Figure 1c and Appendix A, the resulting phase correction is consistent within a series of similar NMR spectra. While this method works very well for series of similar NMR spectra (e.g., from plasma or serum), it does not work if the baseline characteristics of the NMR spectra are very different.

Over the past 30 years, numerous algorithms have been developed for automatic phase correction. Different pieces of software implement different algorithms. To benchmark the quality of the automatic phase correction algorithms, the spectra mentioned above (metabolite mix and *Daphnia* extract NMR spectra) were automatically phase-corrected using MetaboLabPy (version 0.9.66) (in two settings, one where all spectra were automatically phase-corrected and one where the first spectrum was manually phase-corrected, and the other NMR spectra were then automatically phase-corrected comparing the baseline regions at the edges of the NMR spectra), MetaboLab (version 2023.0529; apk5), TopSpin (version 4.4.1; apk0.noe), and the automatic phase correction of MNova (version 15.1.0). Appendix A shows a comparison for the *Daphnia* extract spectra. These spectra were acquired using a 1.7 mm CryoProbe. Panel A plots spectra autocorrected by the MNova software, panel B by TopSpin, panel C by MetaboLabPy using the baseline and reference spectrum-based algorithm, panel D by MetaboLab, and panel E by MetaboLabPy using automatic phase correction for each spectrum. While the general performance of all algorithms is fairly good, TopSpin and the MetaboLabPy algorithm in panel C clearly win the consistency competition. Appendix A shows the same comparison for the metabolite mix. These spectra were acquired using a room-temperature inverse probe. Here, except for the MetaboLab software where the algorithm could not perform an automatic phase correction, the performance is very good for all other pieces of software/algorithms, but again TopSpin and MetaboLabPy (baseline-based algorithm) win the consistency competition, and the spectra phase-corrected by the MetaboLabPy algorithm results in a slightly better baseline condition around the solvent signal.

#### 3.5.2. Further NMR Data Processing Steps

The 1D NMR spectra can be automatically referenced to TMS/TMSP/DSS, to the water chemical shift [13], or manually, e.g., using the anomeric proton of glucose at 5.22 ppm. Post-acquisition water suppression algorithms available include polynomial correction of the FID, convolution with a Gaussian or a sine window, or using the wavelet-based algorithm WAVEWAT [14]. The GUI-based interface for data pre-processing contains all options in an easy-to-use interface, including segmental alignment (Figure 2 and Appendix A). The user graphically selects a region that needs alignment and chooses a reference spectrum. It is vital to select a reference spectrum with a good signal-to-noise ratio in this part of the spectrum. If there is a lack of reference signal in the area, no alignment will occur. The algorithm calculates the optimum shift by using a simple correlation between the current and the reference spectrum. In the example shown, the correlation coefficients between the reference spectrum and the other spectra for the segment shown were generally below 0.3 before alignment and >0.95 after the alignment procedure.

### 3.6. 2D NMR Data Processing and Tracing Analysis

Figure 3 demonstrates the GUI for 2D HSQC multiplet analysis. MetaboLabPy currently has HSQC chemical shift information for over 100 metabolites, with more to be added soon. However, the data are stored in text files (.mlinfo files; see Appendix A for an example), so users can add any metabolites not covered in the MetaboLabPy database. If carbon–carbon scalar coupling (JCC) information is available, the HSQC signal can be peak-picked automatically. The software also calculates a coefficient of determination, indicating the reliability of the analysis. Apart from a percentage value, the numbers are colour-coded, with green indicating good analysis quality, amber indicating some uncertainties and red indicating that this analysis was unsuccessful. The algorithm for peak-picking, multiplet evaluation, and the calculation of the coefficient of determination are described in detail in [10].

The scalar coupling values for central metabolism metabolites are well covered. Each HSQC signal can be selected via a pushbutton, displaying the 2D signal and a 1-dimensional vertical trace, including a fitted line shape of the multiplet (Figure 3a). Percentages, indicating how much each multiplet component contributes to the overall multiplet, are available in the Multiplet Analysis tab (Figure 3b). All analysed multiplet data can be exported to an Excel file, which can then be used to transform the multiplet data into isotopomer distributions (Appendix A). See the MetaboLabPy help system for further details. Progress in signal assignment can be reviewed on the NMR Spectrum tab (Figure 3c).

While there is a particular emphasis on the analysis of tracing data, the software can easily read and process other 2D-NMR datasets, such as TOCSY-, TROSY-, or 2D-Jres-spectra. The latter can then be projected to form 1D-NMR spectra, which can then undergo data pre-processing like normal 1D-NMR spectra.

### 3.7. Performance Comparisons for Batch Processing

All four pieces of processing software offer batch processing capabilities. For TopSpin, a python script, shown in the Appendix A, was used for batch processing. Spectra were dragged and dropped for the MNova software. MetaboLab and MetaboLabPy used their scripting facilities, and both scripts are show in the Appendix A. All scripts (drag and drop for MNova) read spectra in Bruker format, zero filled to 131,072 data points, and apodised the FID with 0.3 Hz line-broadening. Fourier transformed the FIDs and automatically phase-corrected the resulting spectra. To make up a large enough dataset, the four Daphnia extract spectra were copied, resulting in a dataset with 100 NMR spectra. MetaboLabPy is the fastest software using the reference spectrum, the baseline-based algorithm, which takes 28 s for all 100 NMR spectra. Second is the MNova software, which took 45 s. Third is TopSpin, which phase-corrected all 100 NMR spectra in 1 min 13 s. Fourth is MetaboLabPy, with individual automatic phase correction of all 100 NMR spectra in 1 min 24 s. MetaboLab is last, with a total processing time of 3 min 53 s. All spectra were processed on a MacBookPro with macOS 14.7.2.

### 3.8. The Plot Editor

While most NMR data processing/analysis software packages take great care to provide optimised and up-to-date algorithms, data plotting and figure creation are quite often secondary. However, to present the data analysis results, it is often necessary to display NMR spectra or visually represent the quality of data analysis. MetaboLabPy integrates a plot editor, where the user can graphically control all significant aspects of the plots. Axes to be plotted can be selected, as well as aspect ratios of the figures, the spectra and axes’ linewidths, and the font size of axes and labels. The software can be run in light or dark mode. While dark mode is beneficial on computer screens, plotting is usually undertaken for articles or posters, which would require light mode. The plot editor, therefore, enables the user to plot in light or dark mode irrespective of the primary software setting. The goal of the plot editor is to allow the user to produce publication-quality plots without the need for extra editing. An example of HSQC analysis plots is shown in Appendix A.

### 3.9. Use Cases

The main focus of the MetaboLabPy software is metabolomics data processing, as well as analysis of metabolic tracing data. Therefore, two basic use cases are considered:Batch processing and data pre-processing to prepare a series of NMR spectra for uni- or multi-variate statistical data analysis using MetaboAnalyst:
Spectra are read into the software using the scripting interface. MetaboLabPy provides several example scripts. The 1D script to read in the *Daphnia* spectra is shown in the Appendix A. Help for this feature is available here: https://ludwigc.github.io/metabolabpy/scripts.html#scripts.The first spectrum is then phase-corrected, either by typing autophase1d() into the MetaboLabPy command line or by pressing alt + A (option +A on macOS). Should the resulting phase correction not be good enough, the first spectrum can be graphically phase-corrected using the manual phase correction in the Data- > Phase/Baseline Correction- > Interactive Phase Correction menu entry. Help for interactive phase correction is available here: https://ludwigc.github.io/metabolabpy/basicnmrdataprocessing.html#phasecorrection. The other spectra can then phase-corrected automatically using the reference spectrum baseline-based algorithm by typing autophase1d_all_bl() into the MetaboLabPy command line.Data pre-processing options can then be selected graphically using the Data Pre-Processing GUI elements, which includes exclude regions, segmental alignment, noise filtering, bucketing, TSA and PQN normalisation, variance stabilisation, and export. Here, the MetaboAnalyst option generates input files that can be directly imported into MetaboAnalyst. Help for data pre-processing is available here: https://ludwigc.github.io/metabolabpy/basicnmrdataprocessing.html#preprocessing (accessed on 5 January 2024).Analysis of metabolic tracing data:
NMRPipe processed NMR spectra can be read in using one of the example scripts (shown in the Appendix A).The 2D-^1^H ^13^C HSQC NMR spectra can be manually phase-corrected. Instructions for this are shown here: https://ludwigc.github.io/metabolabpy/tracing.html#top (accessed on 5 January 2024).MetaboLabPy carries a database of HSQC chemical shifts for over 100 metabolites and carbon–carbon J coupling information for about 15 metabolites. Users can extend this database as the information is kept in text files, which are easily accessible to users.The GUI then assists in signal assignment and multiplet analysis. Help is available here: https://ludwigc.github.io/metabolabpy/tracing.html#multipletanalysis (accessed on 5 January 2024).The NMR-based multiplet information can then be combined with GC-MS data to convert this information into isotopomer distribution. MetaboLabPy provides Jupyter notebooks to demonstrate how this can be achieved. Help is available here: https://ludwigc.github.io/metabolabpy/tracing.html#isotopomeranalysis (accessed on 5 January 2024).

## 4. Discussion

MetaboLabPy is a new free and open-source software for metabolomics and tracing data analysis. MetaboLabPy is close in scope to the older NMRLab [15] and MetaboLab [11] software. NMRLab is a MATLAB-based package for the general processing of NMR spectra. Later, metabolomics and metabolic tracing specific modules were added to the software, and the name was changed to MetaboLab. While the basic functionality of MetaboLabPy is very similar to that of MetaboLab, MetaboLabPy is actively developed and incorporates advanced algorithms for automatic phase correction, while the automatic phase correction in MetaboLab requires an internal reference peak and is optimised for CryoProbe data, which does not perform well for room-temperature data. While MetaboLabPy provides a signal integration module, where the user can provide a reference integral to quantify signal integrals, the software currently does not offer anything similar to Chenomx’s quantification possibilities.

MetaboLabPy contains an intuitive graphical user interface to help non-expert users process and analyse NMR-based metabolomics and metabolic tracing data. The software features a comprehensive, integrated help and tutorial system, which is also available online.

MetaboLabPy provides facilities for batch-processing large series of NMR spectra and has been used in published metabolomics articles [16,17]. The software can be easily installed using the Anaconda Python distribution (https://www.anaconda.com) and is available on PyPI (https://pypi.org/project/metabolabpy, https://pypi.org/project/qtmetabolabpy, https://pypi.org/project/metabolabpytools, all accessed on 6 January 2025). A tutorial on how to install MetaboLabPy is available here: https://ludwigc.github.io/metabolabpy/softwareinstall.html, accessed on 6 January 2025. The current version is 0.9.66. The software is frequently updated and will always be free and open-source software.

New functionality will be added in the future as needs arise. We are happy to implement new algorithms that users of the software find helpful. A particular emphasis is put on machine learning and deep-learning algorithms. The automated HSQC peak-picking algorithm is an example of a machine-learning approach. We are currently developing a deep-learning-based approach to translate HSQC multiplet information and GC-MS data into isotopomer distributions. This process usually requires biochemical expertise, as presently available technologies do not cover enough data points to calculate isotopomer distributions without manually introduced bias. We hope the deep-learning approach will use indirect information hidden in the data. Isotopomer computation is problematic because the number of independent data points obtainable is generally smaller than the number of fitted isotopomers. This means that the system is underdetermined. As a consequence, the user needs to pre-select which isotopomers are to be fitted. While this process, at least in part, can use biochemical knowledge, it introduces an inherent bias into the analysis and requires that the user is highly proficient in cellular metabolism. The deep-learning approach can use indirect information hidden in the data. Appendix A shows a comparison of a classically fitted isotopomer distribution for lactate vs. the distribution obtained via the deep-learning algorithm. While the classically determined isotopomer distribution was obtained by restricting the fitted isotopomers to be unlabelled and [3-^13^C] and [2,3-^13^C] lactate only, the deep learning algorithm obtains values for all possible eight isotopomers. The two analyses are in very good agreement.

New planned features for 2025 include publication of the deep-learning algorithm to obtain bias-free isotopomer distributions and a new graphical user interface for 1D-NMR quantification using line shape simulations in the style of Chenomx. While the GUI element can be provided fairly easily, the build up of a database will take more time and may depend on future grant funding.

The software will be maintained for at least 10 years, with support lasting for an additional 3 years.

## Figures and Tables

**Figure 1 metabolites-15-00048-f001:**
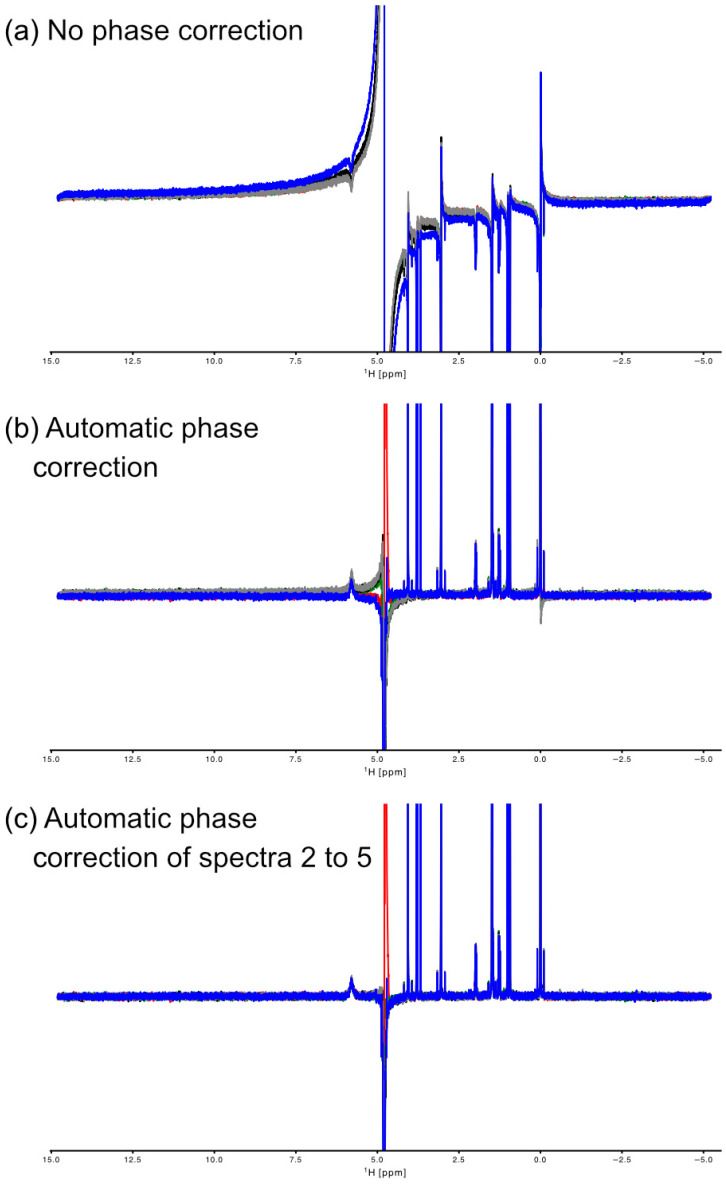
Automatic phase correction of metabolite mixture NMR spectra. The spectra were acquired on a 600 MHz Bruker IVDr equipped with a room-temperature inverse probe. NMR spectra of sample 1 are plotted in blue, from sample 2 in red, from sample 3 in green, from sample 4 in black, and from sample 5 in gray. Panel (**a**) shows the NMR spectra before phase correction. Panel (**b**) depicts the spectra after automatic phase correction using the algorithm described in [12]. The blue spectrum in panel (**c**) was manually phase-corrected; all other spectra were then automatically phase-corrected with the blue spectrum as reference.

**Figure 2 metabolites-15-00048-f002:**
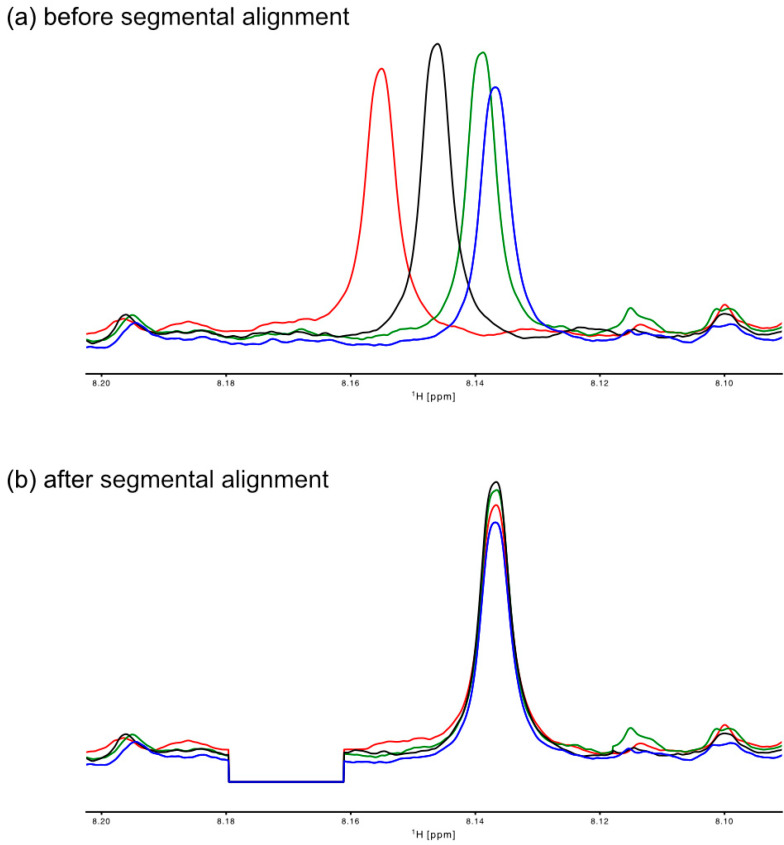
The data pre-processing options in MetaboLabPy include separate alignments for small sections of NMR spectra. NMR spectra from sample 1 are plotted in blue, from sample 2 in red, from sample 3 in green, and from sample 4 in black. The spectra displayed here are from *Daphnia* extracts. The spectra were acquired on a Bruker 600 MHz Avance III system equipped with a 1.7 mm CryoProbe. Panel (**a**) shows the spectra before segmental alignment, and panel (**b**) plots the spectra after alignment. Areas of the spectra shown as zero will be excluded from statistical analysis.

**Figure 3 metabolites-15-00048-f003:**
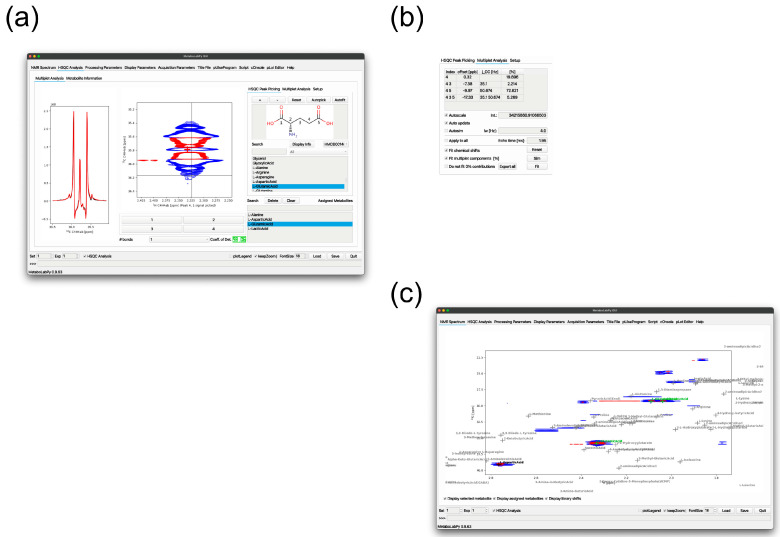
HSQC assignment and analysis tool. Panel (**a**) shows the main interface, where the user can select the metabolite to be analysed. Depending on the number of expected NMR signals in the spectrum, pushbuttons appear below the 2D spectrum display. A trace through the experimental 2D spectrum is shown on the left (black) together with a line shape analysis of the multiplet (red). The GUI element on the right hand side contains several tabs, one of which contains information about the relative multiplet composition, as shown in panel (**b**). Panel (**c**) shows the main spectrum tab, which the user can switch on and off to see which metabolites are already assigned, where all the different resonances of the currently selected metabolites and other resonances are available in the library. Blue contour lines represent positive contours, whereas negative contour lines are shown in red.

## Data Availability

All NMR data presented here are openly available in OSF at http://doi.org/10.17605/OSF.IO/R64G8.

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
