# Peer review of "MetaboLabPy—An Open-Source Software Package for Metabolomics NMR Data Processing and Metabolic Tracer Data Analysis"

_metabolites, 2025, doi:10.3390/metabo15010048_

Round 1

Reviewer 1 Report

Comments and Suggestions for Authors

This manuscript introduces MetaboLabPy, an open-source Python package specialized for processing and analyzing 1D and 2D NMR metabolomics data, and metabolic tracer analysis. This comprehensive software package provides both GUI and non-GUI interfaces and is freely available under a permissive GPL license via PyPI, although still under active development. It is very impressive that such a feature-rich package has been developed by a single author.

Overall, the manuscript is well-written and of high quality. I only have a few minor suggestions:

  1. Broken hyperlinks (lines 301-302): While the hyperlink text may appear correct, the actual links seem to be broken. Please fix each hyperlink and test them by clicking on the links in the generated PDF to ensure they work properly.
  2. Comparison with related software (lines 292-294): The author mentions that MetaboLabPy is similar in scope to older software like NMRLab and MetaboLab. It would be helpful to elaborate on the key similarities and differences between MetaboLabPy and these related packages. For example, the manuscript states that MetaboLabPy can save data in MetaboLab's MATLAB format. Are there other notable commonalities or distinctions?
  3. The manuscript focuses on 2D 1H-13C HSQC spectra. Could the author comment on whether the software can support similar 2D experiments such as TROSY or 1H-15N HSQC?
  4. Excel spreadsheet formats: The author mentions that multiplet and isotopomer analysis results can be exported to Excel files. It would be beneficial to add brief descriptions of the structure and contents of these Excel files. Perhaps as a Table or Figure in the SI. 

Author Response

This manuscript introduces MetaboLabPy, an open-source Python package specialized for processing and analyzing 1D and 2D NMR metabolomics data, and metabolic tracer analysis. This comprehensive software package provides both GUI and non-GUI interfaces and is freely available under a permissive GPL license via PyPI, although still under active development. It is very impressive that such a feature-rich package has been developed by a single author.

Overall, the manuscript is well-written and of high quality. I only have a few minor suggestions:

  1. Broken hyperlinks (lines 301-302): While the hyperlink text may appear correct, the actual links seem to be broken. Please fix each hyperlink and test them by clicking on the links in the generated PDF to ensure they work properly.

Thank you for pointing this out. I have corrected all broken hyperlinks and made sure that all hyperlinks are now working.

  1. Comparison with related software (lines 292-294): The author mentions that MetaboLabPy is similar in scope to older software like NMRLab and MetaboLab. It would be helpful to elaborate on the key similarities and differences between MetaboLabPy and these related packages. For example, the manuscript states that MetaboLabPy can save data in MetaboLab's MATLAB format. Are there other notable commonalities or distinctions?

Thank you for your comment. I added a more elaborate comparison between MetaboLabPy and MetaboLab (NMRLab is just an older version of MetaboLab that did not contain metabolomics and metabolic tracing functionality). I also added benchmarking, comparing MetaboLabPy’s NMR data processing speed with MetaboLab, TopSpin and MNova. While MetaboLabPy can read in MetaboLab’s MATLAB format the software cannot export processed data into that format as the MetaboLab software is considered to have reached end-of-life status.

  1. The manuscript focuses on 2D 1H-13C HSQC spectra. Could the author comment on whether the software can support similar 2D experiments such as TROSY or 1H-15N HSQC?

I added a comment stating, that these 2D-NMR spectra can be read by the software and that it is possible to process these spectra. However, the analytical tools provided are focussed on metabolomics and metabolic tracing analysis.

  1. Excel spreadsheet formats: The author mentions that multiplet and isotopomer analysis results can be exported to Excel files. It would be beneficial to add brief descriptions of the structure and contents of these Excel files. Perhaps as a Table or Figure in the SI.

This has been added to the supplementary material (Figure S7-S9)

Reviewer 2 Report

Comments and Suggestions for Authors

The article presents MetaboLabPy as a promising open-source software package for NMR-based metabolomics and metabolic tracing data analysis. The well-illustrated documentation and instructional videos the authors have provided are commendable. However, there are several aspects of the paper that require revision to better showcase the software's capabilities and improve the overall quality of the manuscript.

1. The abstract should be concise and written as a single paragraph. For software articles, it is essential to include the URL for accessing the software at the end of the abstract.

2. The results section should be expanded to include the following: (a) a detailed description of the software, including its implementation and functionality; (b) an evaluation and comparison with similar software packages; and (c) use cases and examples demonstrating how the software can assist users in their research.

3. Incorporate a workflow diagram to illustrate how MetaboLabPy operates, which will help readers better understand the software's architecture and functionality.

4. Provide a more in-depth comparison of MetaboLabPy's features and capabilities with existing NMR data processing and analysis software. This will highlight the unique advantages of your software and help readers appreciate its novelty and significance.

5. Include a comprehensive validation and benchmarking of the software's performance, focusing on key features such as automatic phase correction, multiplet analysis, and isotopomer distribution calculation. This will demonstrate the robustness and reliability of MetaboLabPy.

6. Incorporate detailed case studies or examples that showcase the software's application in real-world metabolomics and metabolic tracing projects. This will illustrate how MetaboLabPy enables new insights and streamlines data analysis workflows, emphasizing its practical value to the research community.

7. In the discussion section, commit to maintaining and supporting MetaboLabPy for at least five years. This assurance will give users confidence in adopting the software and investing time in learning and integrating it into their research workflows.

Author Response

The article presents MetaboLabPy as a promising open-source software package for NMR-based metabolomics and metabolic tracing data analysis. The well-illustrated documentation and instructional videos the authors have provided are commendable. However, there are several aspects of the paper that require revision to better showcase the software's capabilities and improve the overall quality of the manuscript.

  1. The abstract should be concise and written as a single paragraph. For software articles, it is essential to include the URL for accessing the software at the end of the abstract.

The abstract has been rewritten as a single paragraph. Hyperlinks to access the software have been added to the abstract.

  1. The results section should be expanded to include the following: (a) a detailed description of the software, including its implementation and functionality; (b) an evaluation and comparison with similar software packages; and (c) use cases and examples demonstrating how the software can assist users in their research.

Thank you for pointing this out. I added a more detailed description of the software, including its implementation and functionality. I also added a more detailed comparison with other software packages, namely MetaboLab, MNova, TopSpin and Chenomx. Two use cases (one for metabolomics data analysis and one for metabolic tracing data analysis) have been added with explanations how MetaboLabPy assist users in their research.

  1. Incorporate a workflow diagram to illustrate how MetaboLabPy operates, which will help readers better understand the software's architecture and functionality.

A workflow diagram has been added to illustrate MetaboLabPy’s metabolomics workflow.

  1. Provide a more in-depth comparison of MetaboLabPy's features and capabilities with existing NMR data processing and analysis software. This will highlight the unique advantages of your software and help readers appreciate its novelty and significance.

I added a short paragraph, comparing MetaboLabPy’s features with the older MetaboLab software and also Chenomx.

  1. Include a comprehensive validation and benchmarking of the software's performance, focusing on key features such as automatic phase correction, multiplet analysis, and isotopomer distribution calculation. This will demonstrate the robustness and reliability of MetaboLabPy.

A comparison of batch processing speed has been added comparing MetaboLabPy, MetaboLab, TopSpin and MNova. These softwares are also compared in performance of their automated phase correction algorithms. Multiplet analysis and isotopomer distribution calculation are compared to the older MetaboLab software.

  1. Incorporate detailed case studies or examples that showcase the software's application in real-world metabolomics and metabolic tracing projects. This will illustrate how MetaboLabPy enables new insights and streamlines data analysis workflows, emphasizing its practical value to the research community.

I added a paragraph reporting two published studies that used MetaboLabPy.

  1. In the discussion section, commit to maintaining and supporting MetaboLabPy for at least five years. This assurance will give users confidence in adopting the software and investing time in learning and integrating it into their research workflows.

This assurance has been added.

Reviewer 3 Report

Comments and Suggestions for Authors

The article presents MetaboLabPy, a Python-based open-source software for NMR-data processing and metabolic tracer data analysis, emphasizing its integration of advanced features and its accessibility for users in metabolomics research. Below is a detailed review along with actionable comments for improvement:

Strengths

Comprehensive Workflow: The paper highlights that MetaboLabPy implements a complete workflow, from data preprocessing (e.g., phase correction, scaling, alignment) to integration with downstream statistical tools like MetaboAnalyst, Batman, and Dolphin. This end-to-end solution is a valuable contribution to the field.

2D-NMR and GC-MS Integration: The inclusion of features to process ultra-high-resolution spectra and derive isotopomer distributions through GC-MS data integration adds significant versatility to the software.

User Accessibility: The implementation of a modern GUI using PySide6 and tutorials within the help system ensures the software's usability for both novice and advanced users.

Modular Design: The separation of functionalities into three packages (metabolabpy, qtmetabolabpy, and metabolabpytools) enhances maintainability and modular usage.

Open-Source Availability: Being available on PyPI makes it easily accessible to researchers, promoting adoption and reproducibility.

Limitations and Suggestions for Improvement

1. The manuscript does not include a comparison of MetaboLabPy with existing tools for NMR-data processing (e.g., MNova, Chenomx, or other open-source alternatives). Providing benchmarks in terms of processing speed, accuracy, and usability would strengthen the argument for MetaboLabPy's adoption. Maybe the discussion section can be expanded for more details. The authors may also add limitations or a comparative table of this software model with others available.

2. While the article describes the functionalities, it does not present detailed quantitative evaluations (e.g., alignment performance, or success in isotopomer distribution derivation). Including performance metrics and case studies would substantiate the claims.

3. The mention of a deep-learning approach under development is promising but underexplored. Providing preliminary results or discussing its expected impact on isotopomer distribution analysis would enhance the narrative.

4. Although the GUI is mentioned as user-friendly, details about its customization potential and user feedback from beta testing are absent. Including screenshots and a discussion of feedback would clarify the user experience.

5.The integrated help system and tutorials are appreciated, but the manuscript does not elaborate on their depth and comprehensiveness. Expanding on these aspects, perhaps by describing the scope of workflows covered in the tutorials, would increase clarity.

6. The article briefly mentions future updates like deep-learning integration but does not provide a detailed roadmap or anticipated improvements. Including a roadmap would help users and developers plan long-term usage.

MetaboLabPy has the potential to become a widely used tool in metabolomics research due to its comprehensive features, open-source nature, and focus on integration with existing analysis platforms. However, the manuscript would benefit from additional details on performance, comparative analysis, and user feedback to better support its claims. Expanding the discussion on current limitations and future updates would further strengthen the work.

Author Response

The article presents MetaboLabPy, a Python-based open-source software for NMR-data processing and metabolic tracer data analysis, emphasizing its integration of advanced features and its accessibility for users in metabolomics research. Below is a detailed review along with actionable comments for improvement:

Strengths

Comprehensive Workflow: The paper highlights that MetaboLabPy implements a complete workflow, from data preprocessing (e.g., phase correction, scaling, alignment) to integration with downstream statistical tools like MetaboAnalyst, Batman, and Dolphin. This end-to-end solution is a valuable contribution to the field.

2D-NMR and GC-MS Integration: The inclusion of features to process ultra-high-resolution spectra and derive isotopomer distributions through GC-MS data integration adds significant versatility to the software.

User Accessibility: The implementation of a modern GUI using PySide6 and tutorials within the help system ensures the software's usability for both novice and advanced users.

Modular Design: The separation of functionalities into three packages (metabolabpy, qtmetabolabpy, and metabolabpytools) enhances maintainability and modular usage.

Open-Source Availability: Being available on PyPI makes it easily accessible to researchers, promoting adoption and reproducibility.

Limitations and Suggestions for Improvement

  1. The manuscript does not include a comparison of MetaboLabPy with existing tools for NMR-data processing (e.g., MNova, Chenomx, or other open-source alternatives). Providing benchmarks in terms of processing speed, accuracy, and usability would strengthen the argument for MetaboLabPy's adoption. Maybe the discussion section can be expanded for more details. The authors may also add limitations or a comparative table of this software model with others available.

Thank you for your comment. This has been added.

  1. While the article describes the functionalities, it does not present detailed quantitative evaluations (e.g., alignment performance, or success in isotopomer distribution derivation). Including performance metrics and case studies would substantiate the claims.

This has been added.

  1. The mention of a deep-learning approach under development is promising but underexplored. Providing preliminary results or discussing its expected impact on isotopomer distribution analysis would enhance the narrative.

I added a paragraph discussing how the deep-learning approach benefits isotopomer distribution calculations and how it may help tracing analysis by removing bias via isotopomer pre-selection.

  1. Although the GUI is mentioned as user-friendly, details about its customization potential and user feedback from beta testing are absent. Including screenshots and a discussion of feedback would clarify the user experience.

Unfortunately, there was no official beta test. I added a link to the online tutorials and described in more detail how they benefit the user. These tutorials contain numerous screen shots for every step of data processing, pre-processing, and tracing analysis.

  1. The integrated help system and tutorials are appreciated, but the manuscript does not elaborate on their depth and comprehensiveness. Expanding on these aspects, perhaps by describing the scope of workflows covered in the tutorials, would increase clarity.

This has been added.

  1. The article briefly mentions future updates like deep-learning integration but does not provide a detailed roadmap or anticipated improvements. Including a roadmap would help users and developers plan long-term usage.

A roadmap has been added to the manuscript. I also emphasized the attention paid to user wishes.

MetaboLabPy has the potential to become a widely used tool in metabolomics research due to its comprehensive features, open-source nature, and focus on integration with existing analysis platforms. However, the manuscript would benefit from additional details on performance, comparative analysis, and user feedback to better support its claims. Expanding the discussion on current limitations and future updates would further strengthen the work.

                  Thank you for your comments. I hope you find the revised version enhanced the article.           

Round 2

Reviewer 2 Report

Comments and Suggestions for Authors

I hope the authors continue to maintain MetaboLabPy beyond the publication of the article.

Reviewer 3 Report

Comments and Suggestions for Authors

Thanks for the edits. Good to go now!